# Positioning Information Based High-Speed Communications with Multiple RISs: Doppler Mitigation and Hardware Impairments

**Ke Wang** , **Chan-Tong Lam \*** and **Benjamin K. Ng**

Faculty of Applied Sciences, Macao Polytechnic University, Macao 999078, China; ke.wang@mpu.edu.mo (K.W.); bng@mpu.edu.mo (B.K.N.)
* Correspondence: ctlam@mpu.edu.mo

**Abstract:** In this paper, we consider a multiple reconfigurable intelligent surface (RIS)-assisted system using positioning information (PI) to explore the potential of Doppler effect mitigation and spectral efficiency (SE) enhancement in high-speed communications (HSC) in the presence of hardware impairments (HWI). In particular, we first present a general multi-RIS-assisted system model for HSC with HWI. Then, based on PI, different phase shift optimization strategies are designed and compared for maximizing SE, eliminating Doppler spread, and maintaining a very low delay spread. Moreover, we compare the performance of different numbers of RISs with HWI in terms of SE and delay spread. Finally, we extend our channel model from line-of-sight to the Rician channel to demonstrate the effectiveness and robustness of our proposed scheme. Numerical results reveal that the HWI of RISs increases the delay spread, but has no impact on Doppler shift and spread. Additionally, the multiple RIS system not only suffers a more severe delay spread, but is limited in SE due to the HWI. When the number of RISs increases from 2 to 16, the range of average spectral efficiency and delay spread are from 4 to 4.6 Bit/s/Hz and from 0.7 μs to 2.5 μs, respectively. In contrast to conventional RIS-assisted systems that require channel estimation, the proposed PI-based RIS system offers simplicity without compromising effectiveness and robustness in both SE enhancement and Doppler mitigation.

**Keywords:** multiple reconfigurable intelligent surfaces; hardware impairments; high-speed communications; Doppler effect; spectral efficiency



## 1. Introduction

Due to the evolution of transmissions in high-speed communications (HSC), the low-data rate control signaling system has evolved into diverse data-intensive services [1]. As compared to other wireless networks, HSC systems feature high mobility of the transceivers, along with significant signal penetration losses when passing through the carriages of trains or cars [2]. Consequently, many design difficulties, such as channel modelings, Doppler effect compensations, and time-varying channel estimations, arise as a result of these unique traits [2–5].

Given this background, fifth-generation cellular systems have already incorporated compensating technologies such as massive multiple-input multiple-output (mMIMO) and millimeter waves (mmWave) [1,3,4]. For the upcoming sixth-generation cellular system, both mMIMO and mmWave have been proven inefficient and unsustainable due to high hardware cost, high energy consumption, and high complexity of computation [6,7]. In light of these challenges, recently, reconfigurable intelligent surfaces (RISs) have emerged as a promising technology to leverage these challenges [6–8].

Reconfigurable intelligent surfaces, also known as intelligent reflecting surfaces [8], and large intelligent surfaces [9], have emerged as an innovative technique that enables control of the propagation electromagnetic environment [10,11]. Specifically, ab RIS surface consists of a grid of sub-wavelength passive reflecting elements, which are able to cause

an instantaneous controllable phase and/or amplitude shift on the incoming wave [8]. By doing so, the reflected signal can be reoriented to improve transmission quality [10].

On the other hand, the Doppler effect is a major challenge for HSC systems with dense data sources [2]. Recently, there have already been studies that focused on Doppler mitigation using RIS. Basar [12] discussed the Doppler effect and mitigation in the context of RIS. With the help of RIS, the author investigated the possibility of eliminating or mitigating the multipath fading effect caused by mobile receiver movement. It has been shown that by implementing a real-time tunable RIS, the dramatic changes in signal strength caused by Doppler effects can be mitigated effectively. However, the Doppler shift cannot be compensated for if a direct light-of-sight (LoS) link exists. Björnson et al. [13] introduced a general time-varying model for wireless communication with a single RIS and derived the received signal expression in the complex baseband communication system. Moreover, using the proposed model, they proved that if an uncontrollable direct channel link exists, it is impossible to simultaneously maximize the signal-to-noise ratio (SNR) and compensate for the Doppler shifts introduced by the RIS. This result matches well with that of [12]. Based on the model in [13], Matthiesen et al. [14] utilized predictable information such as mobility patterns to design RIS phase shift sets in advance to reduce the complexity of beamforming.

The above works [12–14] commonly assume perfect RIS hardware and transceivers. Unfortunately, in reality, inherent hardware impairments (HWI) such as phase noises, imperfect channel state information (CSI), and quantization errors, which limit system performances, have to be considered due to the imperfections of the electronic devices [15–18]. In [17], the authors considered the HWI that appears at both RIS and the signal transceivers. Then, closed-form expressions for the average achievable rate (ACR) and the RIS utility were derived; the HWI results in reductions of ACR and RIS utility in more reflecting elements. In [18], the authors analyzed the spectral and energy efficiency of RIS-assisted multiple-input single-output downlink systems with HWI. It is shown that the spectral efficiency (SE) is limited due to HWI even when the numbers of transceiver antennas and RIS elements grow infinitely large, which is in contrast with the conventional case with ideal hardware. Furthermore, we [19] revealed that HWI on the RIS has no impact on Doppler shift and spread, but would increase delay spread.

Note that [17–19] mainly studied single-RIS scenarios. Recently, multiple RISs system has been studied in terms of performance enhancement [20–23]. In [20], the authors formulated and tackled a novel RIS-user/BS association problem that aims to optimally balance the passive beamforming gains from all RISs among different BS-user communication links. In [21], the authors introduced a new RIS-aided communication system, where multiple RISs assist communication between a multi-antenna base station (BS) and a remote single-antenna user by multi-hop signal reflection. In [22], a novel hybrid beamforming scheme for multi-hop RIS-assisted communication is proposed in order to improve the coverage range at THz-band frequencies. Moreover, the authors of [23] studied the statistical characterization and modeling of distributed multi-RIS-assisted communication systems. However, Refs. [20–23] do not consider multiple RISs in HSC systems, especially in terms of Doppler mitigation and HWI analysis.

On the other side, real-time positioning information (PI) is essential to an HSC system to ensure successful operation (e.g., collision prevention) [6,24]. The trajectory of a train or a car can be sensed or predicated using some techniques such as track circuits [24], RF tags [25], odometers [26], global navigation satellite systems [27–30], and smartphones [31,32]. Furthermore, machine learning can also enhance the accuracy of the positioning [32–35]. It is noteworthy that trajectory positioning is different from conventional channel estimation since the PI is only related to times, speeds, and locations. However, there are few papers designing the phase shift set of RIS using PI.

Channel estimation for HSC is always very challenging due to time-varying channels [36]. The estimation process is more costly and inaccurate compared to estimating static/quasi-static channels [37,38]. Therefore, the research gap can be summarized

as follow. First, although there are some works on Doppler mitigation using RIS in HSC, there is little literature considering the multiple RISs case; Second, it is crucial to study the relationship between Doppler mitigation and HWI on an RIS-assisted system, but there are few papers in this direction; Third, SE analysis is also very necessary when considering HWI on a Doppler-mitigation HSC system with the help of RIS; Fourth, how to reduce the computational complexity and energy consumption of channel estimation for RIS-assisted HSC systems is of great importance.

In light of the research gap above, the motivation of this paper is that, instead of using conventional channel estimation techniques, PI (e.g., vehicle positions) could be fully utilized to design the phase shift set in real-time. In particular, the system model proposed in this paper is only related to transmission distance and time; thus, using a well-designed phase shift set, an RIS can reconfigure the reflected phase to align all channels to successfully maximize the received power. The benefit of the proposed scheme is that traditional channel estimations are not needed. Therefore, it can be easily implemented and is more energy-efficient than other conventional RIS-assisted systems.

In this paper, we explore the capacity for Doppler mitigation and received power maximization of applying PI-based multiple RISs with HWI (The simulation results can be reproduced using code available at: https://github.com/ken0225/Multi-RIS-Doppler-Mitigation-Hardware-Impairments, accessed on 8 July 2022). In order to elaborate, in Table 1, we explicitly contrast our contributions to the state-of-the-art RIS-assisted high-speed communications. In particular, Ref. [12] considered RIS with perfect CSI, which is impractical. Moreover, although the multi-RIS scenario is considered, each RIS contains just a few elements in [12], which cannot obtain enough beamforming gains. Ref. [14] does not consider multiple RISs, HWI, or SE analysis, which we already considered in our manuscript. Besides, the channel estimation methods in [37,38] are all traditionally piloted-based. Although these methods may obtain promising channel parameters, the estimation complexity cannot be ignored. To this end, we utilize PI to replace CSI to obtain the phase shift set in real time, which is less computationally demanding. The main contributions of this paper are as follows.

**Table 1.** State-of-the-art RIS-assisted high-speed communications.

| | Consider Multiple RISs | Channel Estimation Method and Complexity. | Consider Hardware Impairments | Consider Delay Spread Minimization | Consider Spectral Efficiency Analysis |
|---|---|---|---|---|---|
| Basar [12] | Yes | Consider perfect CSI; high. | No | No | No |
| Matthiesen, et al. [14] | No | Predictive information; low. | No | Yes | No |
| Sun, et al. [37] | No | Doppler shift adjustment; moderate. | No | No | No |
| Wu, et al. [38] | No | Least squares; moderate. | No | No | No |
| Ours | Yes | PI-based; low. | Yes | Yes | Yes |

- We model the HWI at multiple RISs as random phase shift errors, and the HWI at the transceiver as additive distortion noises. Based on the modeling of the HWI, we then propose a PI-based time-varying system model for multi-RIS-assisted communications. Furthermore, we also generalize our LoS channel model to the Rician case;
- Using the real-time PI, we design and compare different phase shift sets for multiple RISs, and obtain the phase shift set that can maximize SE, remove Doppler spread,

and keep delay spread at a very low level, even with random trajectory errors. We reveal that the RIS HWI can increase delay spread, but has no impact on Doppler shift and spread;

- We compare the performance of different numbers of RISs in terms of SE and delay spread. We reveal that the delay spread grows with the increasing number of RIS. Therefore, the tradeoff between SE and delay spread should be considered when designing a multi-RIS system. We also mathematically derive the closed-form expression of the SE with respect to the proposed system;
- Detailed simulations are provided to validate the effectiveness and robustness of the proposed system in this paper. In particular, the numerical results reveal that our system is still effective in the Rician channel. Furthermore, the PI-based phase shift set is robust even if the random positioning error exists.

The rest of this article is organized as follows. In Section 2, we present a PI-based time-varying multi-RIS-assisted Doppler mitigation communication system model with HWI. Based on the proposed system, in Section 3 we define the expressions for Doppler spread, Doppler shift, and delay spread. In Section 4, we design and compare different phase shift optimization strategies on multiple RISs. In Section 5, we obtain the expression of SE. In Section 6, we provide numerical simulations and discussions. In Section 7, we conclude this paper. In addition, the summary of notations is listed in Table 2.

**Table 2.** Notations.

| Notation | Description | Notation | Description |
|:---:|:---:|:---:|:---:|
| $\boldsymbol{P}$ | Column vector. | $\lfloor \cdot \rfloor$ | Floor function. |
| $\mathcal{U}$ | Uniform distribution. | $\lceil \cdot \rceil$ | Ceil function. |
| $\mathcal{CN}$ | Complex Gaussian distribution. | $\mathbb{Z}$ | Sets of integers. |
| $(\cdot)^{\mathrm{T}}$ | Transpose operation. | $\mathbb{R}$ | Sets of real numbers. |
| $\jmath$ | Imaginary unit. | $\lvert \cdot \rvert$ | Absolute operation. |
| $(x \bmod y)$ | Remainder of the division of $x$ by $y$. | $\lVert \cdot \rVert$ | Euclidean norm operation. |

## 2. System Model

We consider a downlink transmission model where $K$ non-cooperative RISs are deployed to assist the communication from a BS to a high-speed mobile vehicle with a constant speed $v$ meters/second (m/s). There are no information exchanges between any two non-cooperative RISs; therefore, multi-hop signal reflections [20,21] are omitted. All the RISs are identical, and each has $M \times N$ isotropic elements, where $M$ and $N$ are the row and column numbers of the RIS. In addition, consider the far-field scenario; the spacing of any two RISs is large enough that the interference can be neglected. The BS and the vehicle are both equipped with a single isotropic antenna with the positions $\boldsymbol{P}_{\mathrm{BS}}$ and $\boldsymbol{P}_{\mathrm{V}}(t)$, respectively. Figure 1 shows the proposed system with $K = 2$ RISs; each of them is a uniform planar array that is placed on a rectangular grid spaced $dx$ and $dy$ apart in parallel to the $x - y$ plane in a three-dimensional Cartesian coordinate system. The heights of the $i$-th RIS, the BS, and the vehicle are $h_{\mathrm{RIS}i}$, $h_{\mathrm{BS}}$, and $h_{\mathrm{V}}$, respectively. The PI (e.g., real-time positions) is sent to the BS through existing positioning techniques (e.g., global positioning system [28]). The RIS phase shift set is then designed and sent to all RISs. A summary of the notations of system model coefficients is provided in Table 3.

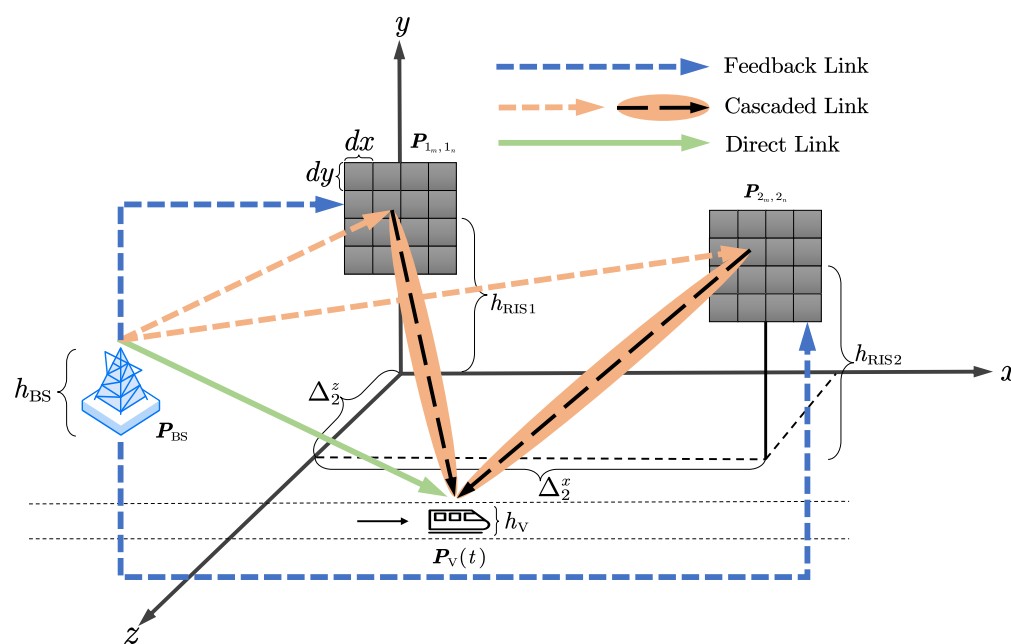

**Figure 1.** A multi-RIS-assisted HSC system. $K = 2$. Note that the real-time phase shift set is sent to RISs through feedback links.

**Table 3.** Notations of system model coefficients.

| Notation | Description | Notation | Description |
|----------|-------------|----------|-------------|
| $P_{BS}$ | BS location. | $d^V_{i_m,i_n}(t)$ | RIS to vehicle distance. |
| $P_V(t)$ | Vehicle location. | $\tau_0(t)$ | Direct link delay. |
| $P_{i_m,i_n}$ | The $(m,n)$-th element of the $i$-th RIS location. | $\tau_{i_m,i_n}$ | Cascaded link delay. |
| $A_0(t)$ | Direct link gain. | $\phi_{i_m,i_n}(t)$ | RIS phase shift set. |
| $A_{i_m,i_n}(t)$ | Cascaded link gain. | $\gamma_{i_m,i_n}$ | RIS HWI. |
| $d_0(t)$ | BS to vehicle distance. | $\eta_t$ | Transmitter HWI. |
| $d^{i_m,i_n}_{BS}$ | BS to RIS distance. | $\eta_R$ | Receiver HWI. |

Therefore, the position column vector of the $(m, n)$-th element on the $i$-th RIS is

$$P_{i_m,i_n} = \left[ \omega(m, dx, M) + \Delta^x_i, \omega(n, dy, N) + h_{\text{RIS}i}, \Delta^z_i \right]^{\text{T}}, \tag{1}$$

where $i = 1, \ldots, K$ and $\omega(m, dx, M) = dx(m - 0.5((M + 1) \bmod 2))$, $m \in \Omega(M)$ with

$$\Omega(\sqrt{M}) = \{ ((\sqrt{M} + 1) \bmod 2)) - \lfloor \sqrt{M}/2 \rfloor, \ldots, \lfloor \sqrt{M}/2 \rfloor \}, \tag{2}$$

and $\omega(n, dy, \sqrt{N})$ is obtained in the same way as $\omega(m, dx, \sqrt{M})$. The $x$ and $z$ coordinates of the center for the $i$-th RIS are $\Delta^x_i \in \mathbb{R}$ and $\Delta^z_i \in \mathbb{R}$, respectively. It is noteworthy that we assume the reflecting sides of all RISs always orient to the vehicle [21].

### 2.1. System Model without Hardware Impairments

Consider that the direct and cascaded channels are all LoS, and the BS, the $(m, n)$-th element of the $i$-th RIS, and the vehicle are located at $P_{BS}$, $P_{i_m,i_n}$, and $P_V(t)$, respectively. Then, the direct link gain is given as [39]

$$A_0(t) = \frac{\lambda_c}{4\pi} \frac{\sqrt{G^V_{BS}(t) G^{BS}_V(t)}}{d_0(t)}, \tag{3}$$

where $G_{\text{BS}}^{\text{V}}(t)$ is the BS antenna gain when the observation point is the vehicle antenna and $G_{\text{V}}^{\text{BS}}(t)$ is the vehicle antenna gain when the observation point is the BS antenna. $\lambda_c$ denotes the wavelength of the signal and $d_0(t) = \|\boldsymbol{P}_{\text{V}}(t) - \boldsymbol{P}_{\text{BS}}\|$ is the Euclidean distance between the BS and the vehicle at time $t$. Since the BS and the vehicle antennas are all isotropic, then

$$A_0(t) = \frac{\lambda_c}{4\pi d_0(t)}. \tag{4}$$

Moreover, the direct link time delay can be obtained as

$$\tau_0(t) = \frac{d_0(t)}{c_0}, \tag{5}$$

where $c_0$ is the speed of light. Therefore, the received signal from direct link at time $t$ can be obtained as

$$S_0(t) = A_0(t)\sqrt{P_t}x(t - \tau_0(t)), \tag{6}$$

where $P_t$ is the fixed transmit power and $x(t)$ is the transmit signal with $\mathbb{E}\{|x(t)|^2\} = 1$. Furthermore, the cascaded link gain by the $(m, n)$-th element of the $i$-th RIS at time $t$ can be obtained as

$$A_{i_m, i_n}(t) = \frac{\lambda_c^2}{16\pi^2} \frac{\sqrt{G_{i_m, i_n}^{\text{BS}} G_{\text{BS}}^{i_m, i_n} G_{i_m, i_n}^{\text{V}}(t) G_{\text{V}}^{i_m, i_n}(t)}}{d_{\text{BS}}^{i_m, i_n} d_{i_m, i_n}^{\text{V}}(t)}, \tag{7}$$

where $G_{i_m, i_n}^{\text{BS}}$ is the gain of the $(m, n)$-th element of the $i$-th RIS when the observation point is the BS antenna, $G_{\text{BS}}^{i_m, i_n}$ is the gain of BS antenna when the observation point is the $(m, n)$-th element of the $i$-th RIS, $G_{i_m, i_n}^{\text{V}}(t)$ is the gain of the $(m, n)$-th element of the $i$-th RIS when the observation point is the vehicle antenna, and $G_{\text{V}}^{i_m, i_n}(t)$ is the gain of the vehicle antenna when the observation point is the $(m, n)$-th element of the $i$-th RIS. Furthermore, $d_{\text{BS}}^{i_m, i_n} = \|\boldsymbol{P}_{i_m, i_n} - \boldsymbol{P}_{\text{BS}}\|$ and $d_{i_m, i_n}^{\text{V}}(t) = \|\boldsymbol{P}_{\text{V}}(t) - \boldsymbol{P}_{i_m, i_n}\|$ are the distances from the BS to the element and from the element to the vehicle, respectively. Since the RISs and transceiver are assumed to be isotropic in this paper, we have

$$A_{i_m, i_n}(t) = \frac{\lambda_c^2}{16\pi^2} \cdot \frac{1}{d_{\text{BS}}^{i_m, i_n} d_{i_m, i_n}^{\text{V}}(t)}. \tag{8}$$

Furthermore, the cascaded link time delay for the $(m, n)$-th element of the $i$-th RIS can be calculated as

$$\tau_{i_m, i_n}(t) = \frac{d_{\text{BS}}^{i_m, i_n} + d_{i_m, i_n}^{\text{V}}(t)}{c_0}. \tag{9}$$

Note that $d_{\text{BS}}^{i_m, i_n}$ is the distance from the BS to the $(m, n)$-th element of the $i$-th RIS. It is fixed, since the BS and all RISs are static. Consequently, the received signal from the cascaded link by the $(m, n)$-th element of the $i$-th RIS can be obtained as

$$S_{i_m, i_n}(t) = A_{i_m, i_n}(t)\sqrt{P_t}x(t - \tau_{i_m, i_n}(t) - \frac{\phi_{i_m, i_n}(t)}{2\pi f_c}), \tag{10}$$

where $\phi_{i_m, i_n}(t) \in [0, 2\pi)$ is the controllable phase shift caused by the $(m, n)$-th element of the $i$-th RIS at time $t$ and $f_c$ is carrier frequency. Note that we consider all the elements as losses diffuse reflectors; thus, the amplitude of $\phi_{i_m, i_n}(t)$ is always 1. Let $\Theta_0(t) \triangleq e^{-j2\pi f_c \tau_0(t)}$ and $\Theta_{i_m, i_n}(t) \triangleq e^{-j2\pi f_c \tau_{i_m, i_n}(t) - j\phi_{i_m, i_n}(t)}$ be the phase parts of the direct and cascaded links, respectively; thus, the corresponding complex baseband received signal without HWI is

$$y(t) = S_0(t)\Theta_0(t) + \sum_{i=1}^{K} \sum_{m=1, n=1}^{M, N} S_{i_m, i_n}(t)\Theta_{i_m, i_n}(t) + w, \tag{11}$$

where $w$ is a white Gaussian noise process with power spectral density $N_0$.

## 2.2. System Model with Hardware Impairments

Based on the model (11), we then obtain the system model expression with HWI. In this paper, two kinds of HWI are considered, i.e., RIS HWI and transceiver HWI. The RIS HWI mainly includes intrinsic hardware imperfections, imperfect CSI, and quantization errors of the RIS hardware [16,17,40,41]. Note that the phase estimation error can be modeled as a zero-mean von Mises variable [41]. However, since the RIS HWI in this paper contains three types of error, we model it as $\gamma_{i_m,i_n} \sim \mathcal{U}[-a_{\mathrm{HWI}}, a_{\mathrm{HWI}}]$ [17], where $a_{\mathrm{HWI}} \in [0, \pi/2]$. It is noteworthy that $\gamma_{i_m,i_n}$ is independent and identically distributed at time $t$. Accordingly, $\Theta_{i_m,i_n}^{\mathrm{HWI}}(t) \triangleq \Theta_{i_m,i_n}(t)e^{-j\gamma_{i_m,i_n}}$. The transceiver HWI includes HWI in the transmitter and receiver. More precisely, the transceiver HWI refers to the distortion noises generated by the transmitter due to inaccurate modeling, which can be modeled as $\eta_t \sim \mathcal{CN}(0, Y_t)$, where $Y_t$ is equal to a proportionality coefficient $\kappa_t$ times transmit power [16,17]. Similarly, the received distortion noises generated by the receiver at time $t$ can also be modeled as $\eta_r \sim \mathcal{CN}(0, V_r)$, where $V_r$ is equal to the proportionality coefficient $\kappa_r$ times received power.

Therefore, the total received signal $y^{\mathrm{HWI}}(t)$ can be obtained as

$$y^{\mathrm{HWI}}(t) = S_0^{\mathrm{HWI}}(t)\Theta_0(t) + \sum_{i=1}^{K}\sum_{m=1,n=1}^{M,N} S_{i_m,i_n}^{\mathrm{HWI}}(t)\Theta_{i_m,i_n}^{\mathrm{HWI}}(t) + \eta_r + w, \tag{12}$$

where

$$S_0^{\mathrm{HWI}}(t) = A_0(t)\left\{ \sqrt{P_t}x(t - \tau_0(t)) + \eta_t \right\}, \tag{13}$$

and

$$S_{i_m,i_n}^{\mathrm{HWI}}(t) = A_{i_m,i_n}(t)\left\{ \sqrt{P_t}x(t - \tau_{i_m,i_n}(t) - \frac{\phi_{i_m,i_n}(t) + \gamma_{i_m,i_n}}{2\pi f_c}) + \eta_t \right\}. \tag{14}$$

It is noteworthy that since the proposed model (12) is only related to the positions of the vehicle at time $t$, which are already known at the BS, we regard the PI as the CSI. Therefore, the distances $d_0(t)$, $d_{\mathrm{BS}}^{i_m,i_n}$ and $d_{i_m,i_n}^{\mathrm{V}}(t)$ are assumed to be already known in the rest of the paper. In addition, we discuss the case when the PI has random errors in Section 6.4.

## 3. Delay Spread, Doppler Shift, and Doppler Spread

In this section, we define delay spread, Doppler shift and Doppler spread for the proposed system. Furthermore, we show that the RIS HWI has no impact on the Doppler shift and Doppler spread, although the delay spread is increased duo to the imperfection of the RIS hardware.

## 3.1. Delay Spread

Recall (5) and (9) at time $t$; the propagation time difference between the $i$-th RIS and direct links is

$$T_{0i}(t) = \max_{i_m,i_n}\left\{ \tau_{i_m,i_n}(t) + \frac{\phi_{i_m,i_n}(t) + \gamma_{i_m,i_n}}{2\pi f_c} \right\} - \tau_0(t), \tag{15}$$

where $\tau_0(t)$ and $\tau_{i_m,i_n}(t)$ are the instantaneous time delays for direct and cascaded links, respectively. $\phi_{i_m,i_n}(t)$ is the phase shift for the $(m, n)$-th element of the $i$-th RI, and $\gamma_{i_m,i_n}$ is the RIS HWI. The difference of propagation time across the $i$-th RIS itself, $T_{\mathrm{RIS}i}(t)$, is defined as

$$T_{\mathrm{RIS}i}(t) = \max_{i_m,i_n}\left\{ \tau_{i_m,i_n}(t) + \frac{\phi_{i_m,i_n}(t) + \gamma_{i_m,i_n}}{2\pi f_c} \right\} - \min_{i_m,i_n}\left\{ \tau_{i_m,i_n}(t) + \frac{\phi_{i_m,i_n}(t) + \gamma_{i_m,i_n}}{2\pi f_c} \right\}. \tag{16}$$

In other words, $T_{\mathrm{RIS}i}(t)$ is the maximum value of the instantaneous delay difference among the total $KMN$ cascaded links. Similarly, the delay spread of the $i$-th RIS and

the direct link is the maximum difference in propagation time overall between the two significant transmission links [14,42], i.e.,

$$T_i(t) = \max\left\{T_{0i}(t), T_{\text{RIS}i}(t)\right\}. \tag{17}$$

Note that $\tau_0 \leq \tau_{i_m,i_n}$; thus, $T_i(t) = T_{0i}(t)$. Therefore, at time $t$, the delay spread for the $K$ RISs is

$$T(t) = \max_i\left\{T_{0i}(t)\right\}, i = 1, 2, \ldots, K. \tag{18}$$

**Remark 1.** *From* (15) *and* (18), *it can be seen that the RIS HWI* $\gamma_{i_m,i_n}$ *increases the delay spread of the whole system. In particular, the upper bound of* $T(t)$ *is*

$$T(t) \leq \max_{i_m,i_n}\left\{\tau_{i_m,i_n}(t) + \frac{\phi_{i_m,i_n}(t)}{2\pi f_c}\right\} - \tau_0(t) + \frac{a_{\text{HWI}}}{2\pi f_c}. \tag{19}$$

*Thus, the delay spread can be maximally increased by* $a_{\text{HWI}}/(2\pi f_c)$. *In other words, the delay spread becomes large when the RIS HWI is severe. However, the transceiver HWI has no impact on delay spread since it is additional thermal noise.*

### 3.2. Doppler Shift

For the proposed system in this paper, the Doppler shift is defined as the difference between the transmitted carrier and the received actual frequencies [42]. Firstly, at time $t$, the Doppler shift for the direct link can be obtained as

$$DS_0(t) = -f_c\frac{d}{dt}\left(\tau_0(t)\right), \tag{20}$$

and the Doppler shift for the cascaded link can be obtained as

$$DS_i(t) = -\max_{i_m,i_n}\left\{\left|DS_{i_m,i_n}(t)\right|\right\}, \quad i = 1, 2, \ldots, K, \tag{21}$$

where $DS_{i_m,i_n}(t)$ is the Doppler shift of the $(m, n)$-th element of the $i$-th RIS, which can be defined as

$$DS_{i_{m,n}}(t) = -f_c\frac{d}{dt}\left(\tau_{i_m,i_n}(t) + \frac{\phi_{i_m,i_n}(t) + \gamma_{i_m,i_n}}{2\pi f_c}\right). \tag{22}$$

Therefore, the Doppler shift for the $K$ RISs is

$$DS(t) = -\max_i\left\{\left|DS_0(t)\right|, \left|DS_i(t)\right|\right\}, \quad i = 1, 2, \ldots, K. \tag{23}$$

**Remark 2.** *Note that* $\mathbb{E}\{\frac{d}{dt}\left(\frac{\gamma_{i_m,i_n}}{2\pi f_c}\right)\} = \frac{d}{dt}\left(\mathbb{E}\{\frac{\gamma_{i_m,i_n}}{2\pi f_c}\}\right) = 0.$ *Then,* (22) *becomes*

$$DS_{i_{m,n}}(t) = -f_c\frac{d}{dt}\left(\tau_{i_m,i_n}(t) + \frac{\phi_{i_m,i_n}(t)}{2\pi f_c}\right). \tag{24}$$

*Consider* (21), (23), *and* (24), *it can be seen that the RIS HWI* $\gamma_{i_m,i_n}$ *has no impact on Doppler shift. Furthermore, similar to delay spread, the transceiver HWI would not affect the Doppler shift either.*

*3.3. Doppler Spread*

The Doppler spread is the maximum difference in instantaneous frequency over all significant propagation links [42]. At time $t$, the Doppler spread between the direct and the $i$-th RIS links is

$$D_{0i}(t) = f_c \max_{i_m,i_n} \left| \frac{d}{dt}\left( \tau_{i_m,i_n}(t) + \frac{\phi_{i_m,i_n}(t) + \gamma_{i_m,i_n}}{2\pi f_c} \right) - \frac{d}{dt}\tau_0(t) \right|, \tag{25}$$

and the Doppler spread across the $i$-th RIS itself, $D_{\mathrm{RIS}i}(t)$, can be obtained as

$$\begin{aligned} D_{\mathrm{RIS}i}(t) =& f_c \max_{i_m,i_n,i_{m'},i_{n'}} \left| \frac{d}{dt}\left( \tau_{i_m,i_n}(t) + \frac{\phi_{i_m,i_n}(t) + \gamma_{i_m,i_n}}{2\pi f_c} \right) \right. \\ & \left. - \frac{d}{dt}\left( \tau_{i_{m'},i_{n'}}(t) + \frac{\phi_{i_{m'},i_{n'}}(t) + \gamma_{i_{m'},i_{n'}}}{2\pi f_c} \right) \right|. \end{aligned} \tag{26}$$

Note that $i_{m'}, i_{n'}$ refers to the different element from the element $i_m, i_n$, in the same RIS $i$. Therefore, for the $i$-th RIS link, at time $t$, the Doppler spread can be obtained as

$$D_i(t) = \max\left\{ D_{0i}(t), D_{\mathrm{RIS}i}(t) \right\}, \quad i = 1, 2, \ldots, K. \tag{27}$$

Consequently, the Doppler spread for the $K$ RISs is

$$D(t) = \max_i\left\{ D_i(t) \right\}, \quad i = 1, 2, \ldots, K. \tag{28}$$

**Remark 3.** *It is noteworthy that, similar to Doppler shift, the RIS and transceiver HWI also have no influence on Doppler spread.*

## 4. Phase Shift Optimization

In this section, we first design the phase shift set to maximize the received power of the proposed system. Then, we discuss the impact of the designed phase shift set on delay spread, Doppler shift, and Doppler spread. We also discuss the deployment strategy for the proposed system.

*4.1. Received Power Maximization*

The optimal phase shift of RIS should be designed to align the phases of direct and cascaded links [8]. Firstly, the $K$ RISs without HWI; then, the optimal phase shift for the $(m, n)$-th element of the $i$-th RIS can be designed as $\phi_{i_m,i_n}(t) = 2\pi f_c\left( \tau_0(t) - \tau_{i_m,i_n}(t) \right) \leq 0$. However, in practical scenarios, the feasible phase range is $[0, 2\pi)$, so we have

$$\phi_{i_m,i_n}(t) = 2\pi f_c\left( \tau_0(t) - \tau_{i_m,i_n}(t) \right) + 2\pi k_{i_m,i_n}(t), \tag{29}$$

where $k_{i_m,i_n}(t)$ is the additional full carrier signal period delay in the $m, n$-th element of the $i$-th RIS at time $t$ [14], which affects the received power. Therefore, maximizing the received power and minimizing the delay spread simultaneously can be transformed into the multi-objective problem as

$$(\mathrm{P1}): \quad \max_{\forall(i_m,i_n):k_{i_m,i_n}(t)} \left[ P_R(t), -T(t) \right], \tag{30}$$

where $P_R(t) = |y^{\mathrm{HWI}}(t) - w|^2$. We will solve P1 in the following section.

*4.2. Delay Spread, Doppler Shift, and Doppler Spread after Phase Shift Optimization*

Based on (29), it is easy to know that if $k_{i_m,i_n}(t) \in \mathbb{Z}$, then $\phi_{i_m,i_n}(t)$ can align the phases of direct and cascaded links. Note that $k_{i_m,i_n}(t) \geq 0$ since $\tau_{i_m,i_n}(t) \geq \tau_0(t)$. Therefore, if



$k_{i_m,i_n}(t) \in \mathbb{Z}_0^+$ and is larger than or equals to $f_c(\tau_{i_m,i_n}(t) - \tau_0(t))$, then the received power can be maximized. $\mathbb{Z}_0^+$ is non-negative integer sets. Accordingly, the Doppler spread between the direct and the *i*-th RIS links is

$$D_{0i}(t) = f_c \max_{i_m,i_n} \left| \frac{d}{dt} \left( \frac{k_{i_m,i_n}(t)}{f_c} + \frac{\gamma_{i_m,i_n}}{2\pi f_c} \right) \right|, \tag{31}$$

where $k_{i_m,i_n}(t) \geq f_c(\tau_{i_m,i_n}(t) - \tau_0(t))$ and $k_{i_m,i_n}(t) \in \mathbb{Z}_0^+$. Based on *Remark 2*, $\frac{d}{dt}\left(\mathbb{E}\{\gamma_{i_m,i_n}\}\right) = 0$, and $\frac{d}{dt}\left(k_{i_m,i_n}(t)\right) = 0$. Then, $\mathbb{E}\{D_{0i}(t)\} = 0$. Similarly, $\mathbb{E}\{D_{\mathrm{RIS}i}(t)\} = 0$ and $\mathbb{E}\{D_i(t)\} = 0$. Therefore, the Doppler spread of the proposed system with *K* RISs with HWI for one vehicle pass is still zero, i.e., $D(t) = 0$.

Considering (17) and (29), we have

$$T_i(t) = \max_{i_m,i_n} \left\{ \tau_{i_m,i_n}(t) + \frac{k_{i_m,i_n}(t)}{f_c} + \frac{\gamma_{i_m,i_n}}{2\pi f_c} \right\} - \tau_0(t). \tag{32}$$

It can be seen that $T_i(t)$ can be affected by $k_{i_m,i_n}(t)$ and $\gamma_{i_m,i_n}$. In other words, the phase shift set and the RIS HWI can increase delay spread. In order to minimize $T(t) = \max\{T_i(t)\}$, $k_{i_m,i_n}(t)$ should be as small as possible. Therefore, the phase shift set of the *i*-th RIS, i.e., the solution of P1 in (30), should be designed as

$$\phi_{i_m,i_n}^{\mathrm{opt}}(t) = 2\pi \left( f_c\left(\tau_0(t) - \tau_{i_m,i_n}(t)\right) + k_{i_m,i_n}^{\mathrm{opt}}(t) \right), \tag{33}$$

where $k_{i_m,i_n}^{\mathrm{opt}}(t) = \lceil f_c(\tau_{i_m,i_n}(t) - \tau_0(t)) \rceil$.

Table 4 compares different $k_{i_m,i_n}(t)$ strategies and shows that $\phi_{i_m,i_n}^{\mathrm{opt}}(t)$ in (33) is a promising design of for a phase shift set, although it cannot mitigate the delay spread due to the existence of the direct LoS link. In particular, Strategy I cannot maximize the received power, since the phase shift set with this strategy is not able to align the phases of direct and cascaded links. The phase shift sets with Strategy II and III can maximize the received power and remove the Doppler spread. However, Strategy II cannot minimize the delay spread, since $k_{i_m,i_n}(t)$ has not been minimized.

**Table 4.** Comparison between different $k_{i_m,i_n}(t)$ strategies.

| Strategy | Result |
|:---:|:---:|
| Strategy I: $k_{i_m,i_n}(t) \in \mathbb{R}$, and $k_{i_m,i_n}(t) > \lceil f_c(\tau_{i_m,i_n}(t) - \tau_0(t)) \rceil$. | Cannot $\max\{P_R(t)\}$. |
| Strategy II: $k_{i_m,i_n}(t) \in \mathbb{Z}$, and $k_{i_m,i_n}(t) > \lceil f_c(\tau_{i_m,i_n}(t) - \tau_0(t)) \rceil$. | $\max\{P_R(t)\}$, remove $D(t)$. |
| Strategy III: $k_{i_m,i_n}(t) = \lceil f_c(\tau_{i_m,i_n}(t) - \tau_0(t)) \rceil$. | $\max\{P_R(t)\}$, remove $D(t)$, and $\min\{T(t)\}$. |

Algorithm 1 generates the phase shift set $\phi_{i_m,i_n}^{\mathrm{opt}}(t)$ using PI. Specifically, PI can be used to calculate the distances $d_0(t)$, $d_{\mathrm{BS}}^{i_m,i_n}$ and $d_{i_m,i_n}^{\mathrm{V}}(t)$ in real time. Then, the time delay $\tau_0(t)$ and $\tau_{i_m,i_n}(t)$ can be obtained. The phase shift set $\phi_{i_m,i_n}^{\mathrm{opt}}(t)$ is thus achieved accordingly. It is noteworthy that this algorithm does not include complicated operations such as iterations or alternative optimizations. Therefore, the proposed system has lower computational complexity.

---

**Algorithm 1:** Phase shift set $\phi_{i_m,i_n}^{\text{opt}}(t)$ generation in (33) for $K$ RISs

---

**Input**: Positioning information $P_{BS}$, $P_{i_m,i_n}$, and $P_V(t)$; total time for one vehicle pass $T$, total RIS number $K$, and element number of each RIS $MN$;

**Initialization**: $\phi_{i_m,i_n}^{\text{opt}}(t) \leftarrow 0$;

**for** $t = 1 : T$ **do**

    Calculate $d_0(t)$ and $d_{\text{BS}}^{i_m,i_n}$ using PI;

    Calculate $\tau_0(t)$ using (5);

    **for** $i = 1 : K$ **do**

        **for** $(m, n) = 1 : MN$ **do**

            Calculate $d_{i_m,i_n}^{\text{V}}(t)$ using PI;

            Calculate $\tau_{i_m,i_n}(t)$ using (9);

            Calculate $\phi_{i_m,i_n}^{\text{opt}}(t)$ using (33);

        **end**

    **end**

**end**

---

Substituting (33) into (22), the Doppler shift of the $(m, n)$-th element of the $i$-th RIS after RIS phase optimization is

$$DS_{i_{m,n}}(t) = -f_c \frac{d}{dt}\left(\tau_0(t)\right).\tag{34}$$

Note that $DS_{i_{m,n}}(t) \neq 0$, since $\tau_0(t)$ exists. In other words, the proposed system can remove the Doppler spread, but the Doppler shift $DS(t)$ still exists. However, if the direct link is blocked, then $DS(t) = 0$. This is so called *Doppler cloaking* [13].

*4.3. Delay Spread Minimization in Multi-RIS Systems*

The delay spread is mitigated from the perspective of designing the phase shift set of the RISs in (33). However, we can also minimize $T(t)$ further through the deployment of the multiple RISs. Consider the definition of delay spread in (18). $T(t)$ can be minimized if the distance of cascaded link is minimized. In order to prevent mutual interference from the RISs themselves, the distance between any two RISs cannot be too small. Therefore, the cascaded link distance of the multiple RISs is often longer than that of the single-RIS case. This implies that a system with $K$ RISs often has a higher delay spread than the case with a single RIS near the BS.

Furthermore, when an extra RIS is added to the multi-RIS system, we have to design a deployment strategy so that the new coming RIS does not increase the delay spread. The following proposition will provide the deployment guidelines.

**Proposition 1.** *The delay spread of the system with K RISs depends only on the RIS that is the farthest from the BS.*

**Proof of Proposition 1.** See Appendix A. □

**Remark 4.** *For the proposed system, let d denotes the distance between the BS and the farthest RIS; then, the system delay spread does not change when an extra RIS is added to the system if the distance between the new RIS and the BS is smaller or equal to d. In other words, the delay spread $T(t)$ can be mitigated by minimizing $d_{\text{BS}}^{i_m,i_n}$. Another way to mitigate delay spread is minimize $d_{i_m,i_n}^{\text{V}}(t)$. However, putting the RIS on the road side is more convenient, compared to deploying it on the vehicle.*

## 5. Spectral Efficiency Analysis

In this section, we analyze the downlink SE for the proposed system using the phase shift set $\phi_{i_m,i_n}^{\mathrm{opt}}(t)$ in (33), considering that RIS and transceiver HWI exists. The result will reveal that the RIS and transceiver HWI can decrease SE in different ways, and as the number of RIS $K$ and transmit power $P_t$ becomes large, the main impairment factor is transceiver HWI instead of RIS HWI.

Recall (6) and (14), and denote the received signal as

$$z(t) \triangleq S_0^{\mathrm{HWI}}(t)\Theta_0(t) + \sum_{i=1}^{K} \sum_{m=1,n=1}^{M,N} S_{i_m,i_n}^{\mathrm{HWI}}(t)\Theta_{i_m,i_n}^{\mathrm{HWI}}(t). \tag{35}$$

Then, the spectral efficiency $\mathrm{SE}(t)$ at time $t$ can be obtained as

$$\mathrm{SE}(t) = \mathbb{E}\left\{ \log_2 \left\{ 1 + \frac{|z(t)|^2}{(\kappa_t + \kappa_r)|z(t)|^2 + N_0} \right\} \right\}. \tag{36}$$

By substituting the optimal phase shift set $\phi_{i_m,i_n}^{\mathrm{opt}}(t)$ in (33) into (36), we characterize the impact of the HWI on the downlink SE, given in the following proposition.

**Proposition 2.** *Consider a system with $K$ identical RISs. Each of them uses the optimal phase shift set $\phi_{i_m,i_n}^{\mathrm{opt}}(t)$ in (33), the RIS HWI $\gamma_{i_m,i_n} \sim \mathcal{U}[-a_{\mathrm{HWI}}, a_{\mathrm{HWI}}]$, where $a_{\mathrm{HWI}} \in [0, \pi/2]$, and the transceiver HWI $\eta_t$ and $\eta_r$. Then, at time $t$, the downlink SE of the proposed system with HWI can be obtained as*

$$\mathrm{SE}(t) \xrightarrow{a.s.} \log_2 \left\{ 1 + \frac{\mathcal{Q}(t)}{(\kappa_t + \kappa_r)\mathcal{Q}(t) + N_0} \right\}, \tag{37}$$

*where $\mathcal{Q}(t)$ is the received power that can be obtained as*

$$\mathcal{Q}(t) = \mathbb{E}\{|z(t)|^2\} \xrightarrow{a.s.} P_t \left\{ A_0^2(t) + \mathrm{sinc}^2(a_{\mathrm{HWI}})A^\star(t) \right. \tag{38}$$

$$\left. + \sum_{i=1}^{K} \sum_{m=1,n=1}^{M,N} A_{i_m,i_n}^2(t) + 2A_0(t)\,\mathrm{sinc}(a_{\mathrm{HWI}}) \sum_{i=1}^{K} \sum_{m=1,n=1}^{M,N} A_{i_m,i_n}(t) \right\},$$

*where*

$$A^\star(t) = A_1(t) \sum_{k \neq 1}^{KMN} A_k(t) + A_2(t) \sum_{k \neq 2}^{KMN} A_k(t) + \cdots + A_{KMN}(t) \sum_{k \neq KMN}^{KMN} A_k(t), \tag{39}$$

*and $k = 1, 2, \ldots, KMN$.*

**Proof of Proposition 2.** See Appendix B. □

*Special Case 1*: For an ideal multi-RIS communication system without any HWI, $\kappa_t = \kappa_r = 0$ and $a_{\mathrm{HWI}} = 0$. Thus, the downlink spectral efficiency $\mathrm{SE}(t)$ reduces to

$$\mathrm{SE}(t) = \log_2 \left\{ 1 + \frac{\mathcal{Q}_{\mathrm{ideal}}(t)}{N_0} \right\}, \tag{40}$$

where

$$\mathcal{Q}_{\mathrm{ideal}}(t) = P_t \left\{ A_0^2(t) + M^2 N^2 A^\star(t) + 2MNA_0 \sum_{k=1}^{KMN} A_k \right\}. \tag{41}$$

**Remark 5.** *Mathematically, the RIS HWI $\gamma_{i_m,i_n}$ leads to two extra scalable factors, $\text{sinc}(a_{\text{HWI}})$ and $\text{sinc}^2(a_{\text{HWI}})$, to the received power $\mathcal{Q}(t)$. Note that $\text{sinc}(a_{\text{HWI}})$ is a monotonically decreasing function when $a_{\text{HWI}} \in [0, \pi/2]$.*

*Special Case 2*: If the total number of RISs is large enough, i.e., $K \gg 1$, then the downlink SE can be rewritten as

$$\text{SE}(t)\big|_{K \gg 1} \xrightarrow{a.s.} \log_2\left\{1 + \frac{1}{\kappa_t + \kappa_r}\right\}. \tag{42}$$

**Remark 6.** *This result reveals that the only factor that can influence SE is the HWI in transceivers when the proposed system has enough RISs. Furthermore, if the transmit power is high, the SE saturates, and (42) is also the upper bound of SE for a high $P_t$ regime. In other words, the SE cannot be further enhanced by increasing K or by increasing the transmit power $P_t$. Besides, increasing K would cause a severe delay spread.*

*Special Case 3*: If all the elements of the $K$ RISs have the same gain $A_1$, then we have

$$\text{SE}_{A_1}(t) \xrightarrow{a.s.} \log_2\left\{1 + \frac{\mathcal{Q}_{A_1}(t)}{(\kappa_t + \kappa_r)\mathcal{Q}_{A_1}(t) + N_0}\right\}, \tag{43}$$

where

$$\mathcal{Q}_{A_1}(t) \xrightarrow{a.s.} P_t\left\{A_0^2(t) + (KMN)^2 A_1^2(t)\, \text{sinc}^2(a_{\text{HWI}})\right. \tag{44}$$
$$\left. + 2KMN A_0(t) A_1(t)\, \text{sinc}(a_{\text{HWI}})\right\}.$$

**Remark 7.** *$A_0(t) = 0$ when the direct link does not exist. Therefore, the total received power $(KMN)^2 A_1^2(t)\, \text{sinc}^2(a_{\text{HWI}})$ is proportional to the total number of the elements $(KMN)^2$. This result matches with the square power law in [8].*

## 6. Simulation Results and Discussions

In this section, numerical evaluations are provided to validate the results in Sections 3–5. As shown in Figure 2, assume the BS is located at $\boldsymbol{P}_{\text{BS}} = [0\text{ m}, h_{\text{BS}}\text{ m}, 200\text{ m}]^{\text{T}}$ and the vehicle is located at $\boldsymbol{P}_{\text{V}}(t) = [x_{\text{start}} + v \cdot t\text{ m}, h_{\text{V}}\text{ m}, 100\text{ m}]^{\text{T}}$ where $t \in \mathbb{Z}_0^+$ and its range is from 1 s to 10 s. Without loss of generality, we can consider all the RISs with zero $z$ coordinate, i.e., $\Delta_i^z = 0$. The simulations are averaged over $10^3$ different realizations. Note that $d_x = d_y = \lambda/(2\sqrt{\pi})$. This is because all the RISs are modeled as lossless diffuse reflectors with isotropic antenna gain, and we ignore the correlations between the elements of each RIS. Parameter settings are shown in Table 5. Unless otherwise specified, all parameters are set to the values in Table 5 by default. All results are obtained by using a personal computer with a 1.8 GHz i7-8565U CPU and 8 GB RAM. The software environment is MATLAB2021a.

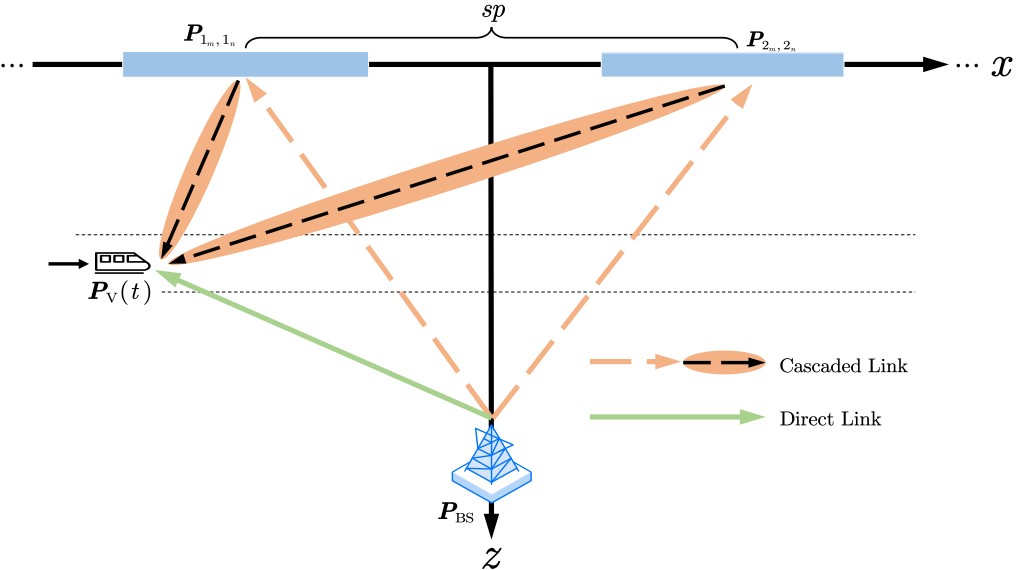

**Figure 2.** Top view of the multi-RIS system considered in this section.

**Table 5.** Parameter settings [19,43].

| Parameter | Value |
|---|---|
| BS height ($h_{\mathrm{BS}}$) | 20 m |
| RIS height ($h_{\mathrm{RIS}}$) | 15 m |
| Vehicle height ($h_{\mathrm{V}}$) | 2 m |
| Speed of the vehicle ($v$) | 100 m/s |
| Start point of the vehicle ($x_{\mathrm{start}}$) | $-500$ m |
| Total time for one vehicle pass ($T$) | 10 s |
| Transmit power ($P_t$) | 20 dBm |
| Power spectral density ($N_0$) | $-80$ dBm |
| Carrier frequency ($f_c$) | 2.4 GHz |
| Rectangular grid spacing of the RIS ($dx$, $dy$) | $\lambda/(2\sqrt{\pi})$, $\lambda/(2\sqrt{\pi})$ |
| The number of each RIS ($M \times N$) | $32^2$ |
| Spacing ($sp$) | 50 m |
| RIS HWI ($a_{\mathrm{HWI}} = \pi/4$) | $\pi/4$ |
| Transceiver HWI ($\kappa_t$, $\kappa_r$) | $0.03^2$, $0.03^2$ |

### 6.1. Different Phase Shift Sets

Figure 3 demonstrates the SE for the multi-RIS system using different phase shift sets for one vehicle pass, which is intended to validate Table 4. Considering $K = 8$, it can be seen that using optimized phase shift set $\phi^{\mathrm{opt}}_{i_m,i_n}(t)$ in (33) achieves a higher SE compared to the case without RIS. For example, the SE is from 6.2 to 6.8 bit/s/Hz when the moving distance increases from 400 m to 480 m. When the $k_{i_m,i_n}$ in (29) is larger than $k^{\mathrm{opt}}_{i_m,i_n}(t)$, e.g., $k^{\mathrm{opt}}_{i_m,i_n}(t) + 1$, then the same value of SE can also be obtained. This matches with the result in Table 4. However, when $k_{i_m,i_n}(t) \in \mathbb{R}$ (e.g., $k^{\mathrm{opt}}_{i_m,i_n}(t) + 0.2$), the SE cannot be maximized, as we expected. In addition, the highest SEs are obtained as 6.9 bit/s/Hz when the moving distance is 500 m, due to the smallest distance $d_0(t)$ in (4) at this point.

Figure 4 shows that the optimized phase shift set $\phi^{\mathrm{opt}}_{i_m,i_n}(t)$ results in a lower delay spread. It can be observed that the delay spread first decreases and then increases, from $1.23 \times 10^{-6}$ s to $1.47 \times 10^{-6}$ s. This is because the vehicle first approaches and then moves away from the BS. If we change the location of the BS or the RISs, the changing pattern of the delay spread will be different. However, using $\phi^{\mathrm{opt}}_{i_m,i_n}(t)$ in (33) can always obtain a lower delay spread. Moreover, the imperfect $k_{i_m,i_n}(t)$ (e.g., $k^{\mathrm{opt}}_{i_m,i_n}(t) + 1$) and the RIS HWI $\gamma_{i_m,i_n}$ will increase delay spread. It is noteworthy that the imperfect phase shift set is possible to

achieve in practice, since the RIS hardware can cause phases over $2\pi$ [14]. The RIS without HWI case can be considered the same as the RIS with perfect CSI in [12,14]. Therefore, compared to the models in [12,14], our analysis is more practical, since HWI is considered in the system model.

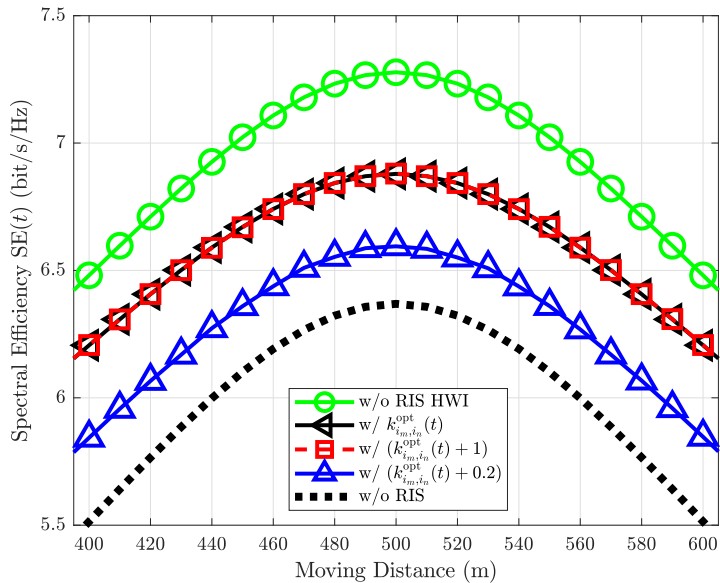

**Figure 3.** Spectral efficiency for the system using different phase shift sets. $K = 8$.

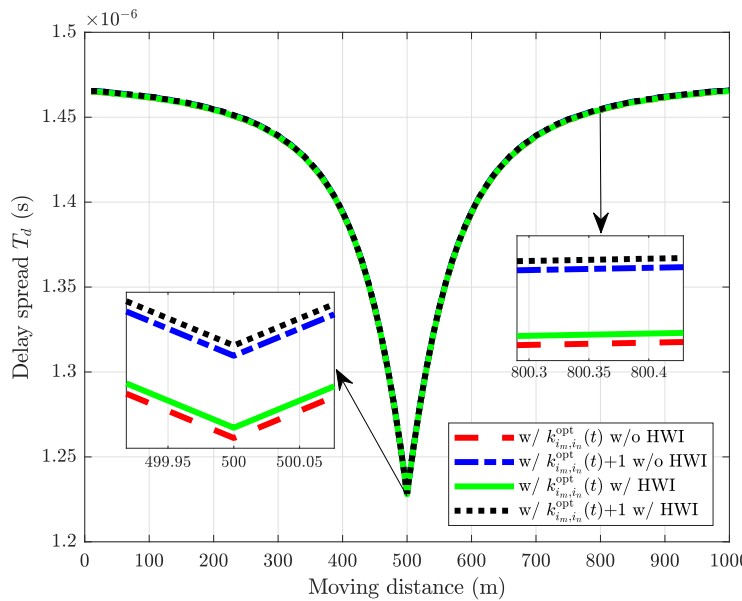

**Figure 4.** Delay spread for the system using different phase shift sets. $K = 8$.

### 6.2. Different Numbers of RIS

Figure 5 shows the SE for different numbers of RISs for one vehicle pass. It can be seen that the SE increases when the number of RIS $K$ becomes larger. More precisely, when the moving distance is 500 m, the SE gap between $K = 2$ and $K = 8$ is 0.6 bit/s/Hz. Interestingly, however, the growth rate of SE is slowed when $K$ is large. In fact, the SE cannot keep increasing even if $K$ becomes infinity, since transceiver HWI exists. Moreover, the analytical expression of SE in (37) matches well with the simulation results. Thus, the figure validates Proposition 2 and we use (37) to generate the rest of figures. It is

noteworthy that Figure 5 also reveals that the multi-RIS scheme proposed in this paper is more promising than the single RIS system in [14], i.e., the $K = 1$ case in the figure.

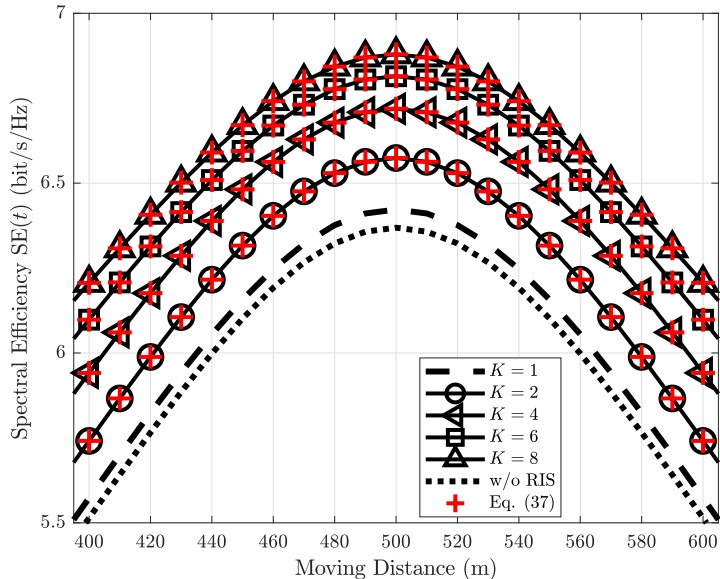

**Figure 5.** Spectral efficiency for different numbers of RIS. $K = 1, 2, 4, 6, 8$. Note that the analytical results are based on (37).

Figure 6 shows the delay spread for different numbers of RISs for one vehicle pass. In particular, more RISs cause a larger delay spread, although the SE can be improved when $K$ is not very large. For example, when the moving distance is 500 m, the $K = 2$ case has $0.7 \times 10^{-6}$ s delay spread, but it increases to $1.2 \times 10^{-6}$ s when $K = 8$. It is noteworthy that the maximum cyclic prefix of 5G is $4.7 \times 10^{-6}$ s [43]. Therefore, if $K$ is large enough, it is possible that the delay spread exceeds the maximum cyclic prefix.

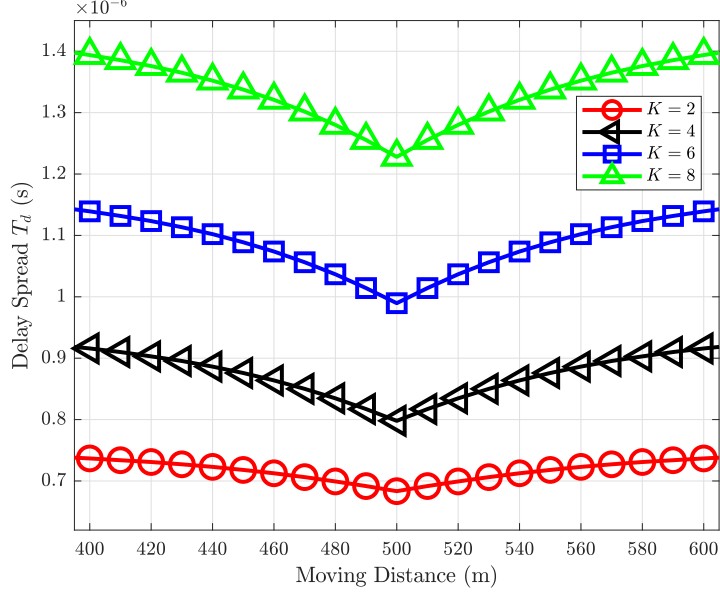

**Figure 6.** Delay spread for different numbers of RIS. $K = 2, 4, 6, 8$.

Figure 7 validates Proposition 1. In particular, considering the moving distance equal to 500 m, the delay spread for the case when $K = 4$ and $sp = 50$ m is the same (i.e., $0.8 \times 10^{-6}$ s) as in the case when $K = 8$ and $sp = 150/7$ m. In other words, the delay spread depends on the RIS that is the furthest from the BS. Similarly, the case of $sp = 100$ m has the same delay spread (i.e., $1.1 \times 10^{-6}$ s) as that of $sp = 300/7$ m if they have 4 and 8 RISs, respectively. Therefore, the delay spread is not be increased if the $K$ RIS are deployed properly. Besides, when the BS and the RIS(s) are all far from the vehicle, the delay spread will be static. This can be observed when the moving distance is 0 m to 100 m and from 900 m to 1000 m. Therefore, with regard to mitigating delay spread, the single RIS next to the BS is the better deployment strategy, although the SE cannot be further improved.

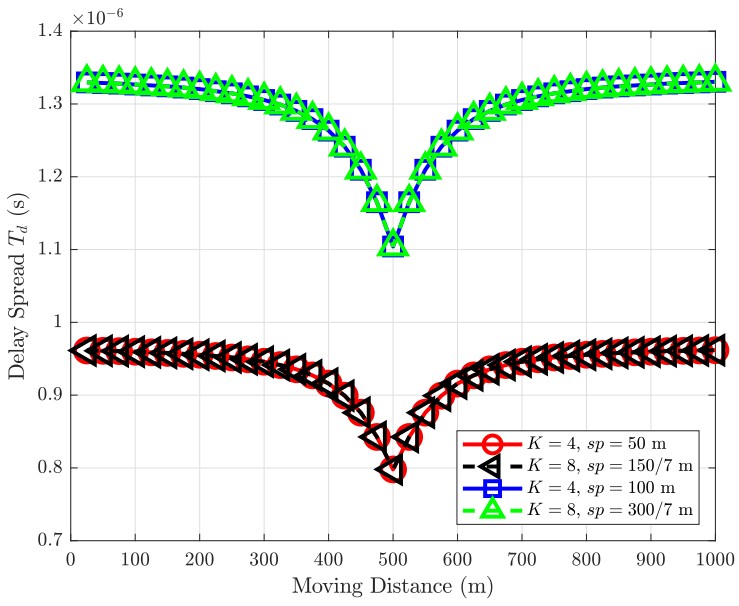

**Figure 7.** Delay spread for different spacings of RIS. $K = 4, 8$.

### 6.3. Different Hardware Impairments

Figure 8 reveals that the impact of RIS HWI diminishes in high SNR regions. In particular, when the moving distance is from 400 m to 600 m, i.e., the system is in a high SNR region, then the HWI in transceivers mainly decreases the SE instead of the RIS HWI. For example, the case $a = \pi/6$, $\kappa_t = \kappa_r = 0.07^2$ performs the worst with the high SNR, but after 700 m, its performance is the same as $a = \pi/2$, $\kappa_t = \kappa_r = 0.03^2$. In other words, after 700 m, the main factor that decreases the SE is the RIS HWI. Besides, in high SNR regions, the SEs of the two cases with $\kappa_t = \kappa_r = 0.03^2$ are higher than the case with $\kappa_t = \kappa_r = 0.07^2$. However, if the SNR becomes lower (e.g., the moving distance larger than 600 m), the impact of the RIS HWI becomes severe.

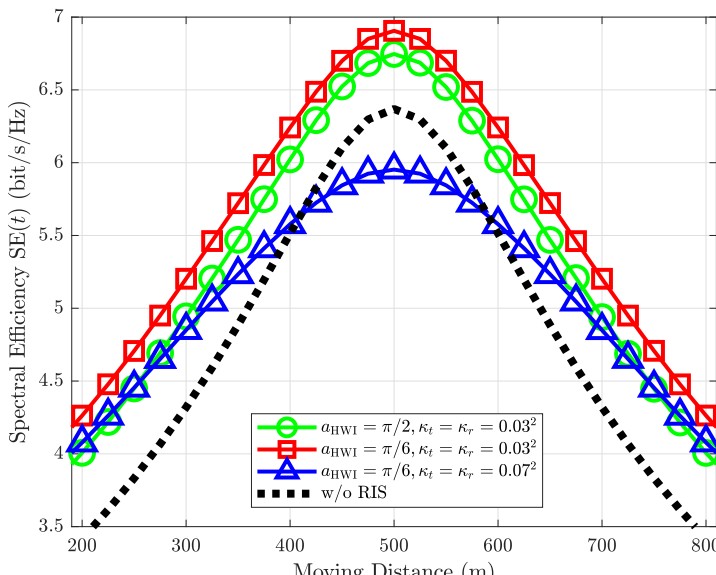

**Figure 8.** Spectral efficiency of the multi-RIS system with different HWI for one vehicle pass. $K = 8$.

Figures 9 and 10 validate *Special Case 2*. The figures show the relationship between the highest SE for one vehicle pass and different $K$ ($P_t$) when the optimal phase shift set (33) is used. It can be seen that when $K$ ($P_t$) becomes large, the main factor that decreases the SE is transceiver HWI, i.e., $\kappa_t$ and $\kappa_r$. For example, the SE grows significantly from 3.5 bit/s/Hz to 6.5 bit/s/Hz when the transmit power $P_t$ is from 0 dBm to 20 dBm. When $P_t = 30$ dBm, however, there is almost no SE increment. The same pattern can also be observed when $K$ increases from 2 to 16. Besides, the RIS HWI has minimal impact on SE when $P_t = 30$ dBm or $K = 16$.

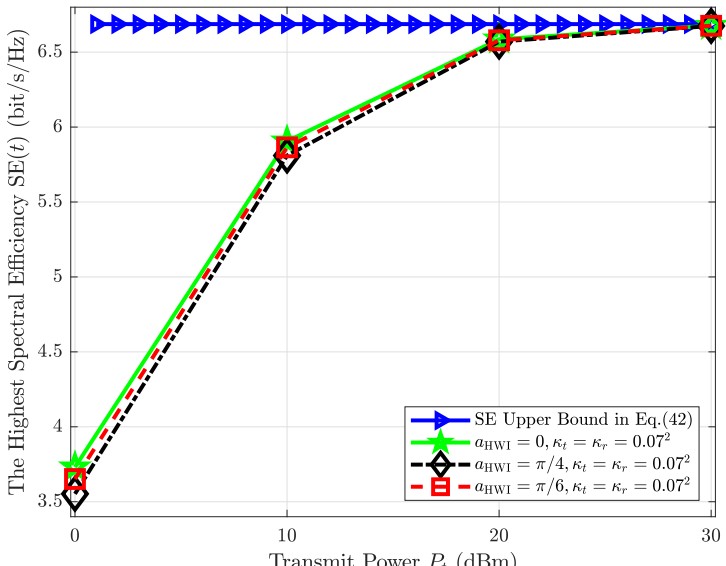

**Figure 9.** The highest spectral efficiency for one vehicle pass with different transmit power $P_t$. $K = 4$, $M \times N = 128^2$, and $P_t = 0, 10, 20, 30$ dBm. Note that the upper bound for SE is based on (42).

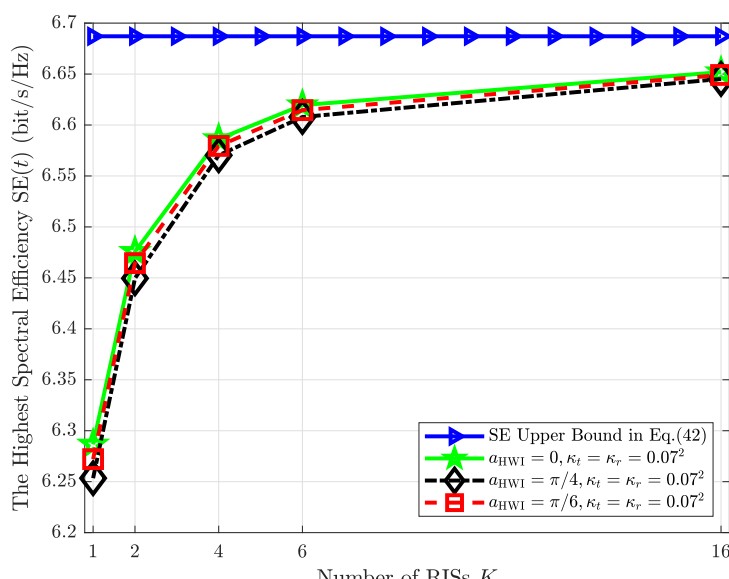

**Figure 10.** The highest spectral efficiency for one vehicle pass with different numbers of RISs $K$. $K = 2, 4, 6, 16$, and $M \times N = 128^2$. Note that the upper bound for SE is based on (42).

### 6.4. Random Errors in Trajectory

The above works are all under the assumption that positioning information is perfect. In practical scenarios, however, the trajectory may have some random errors [24,25,27,28]. Considering that the random error follows a uniform distribution, then the position of the vehicle becomes $\boldsymbol{P}_V(t) = [x_{\text{start}} + v \cdot t + \gamma_{\text{traj}} \text{ m}, h_V \text{ m}, 100 + \gamma_{\text{traj}} \text{ m}]$, where $\gamma_{\text{traj}} \sim \mathcal{U}[-a_{\text{traj}}, a_{\text{traj}}]$. Figure 11 shows the spectral efficiency for one vehicle pass considering that random trajectory errors exist. It is shown that the performances of all random error cases are almost the same. Therefore, the proposed system is robust to positioning information error. Moreover, it is noteworthy that compared with [37,38], using PI for obtaining CSI is more energy-efficient since conventional pilot-based methods are not necessary.

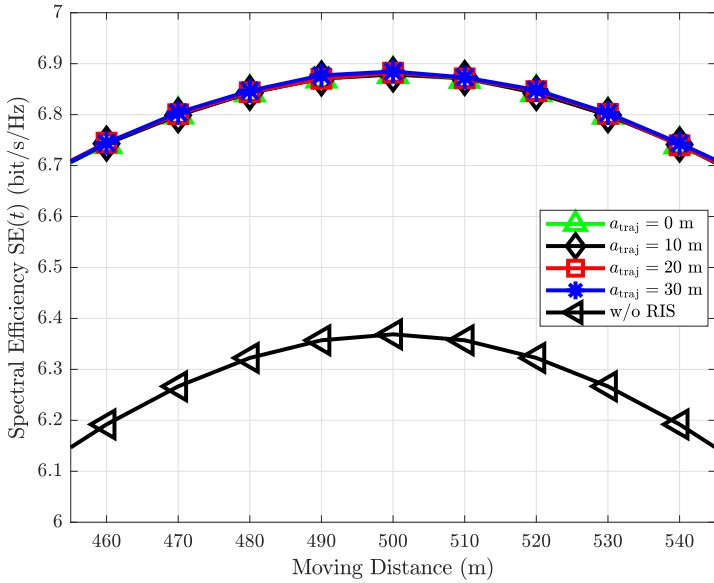

**Figure 11.** Spectral efficiency for one vehicle pass with random errors in trajectory. $K = 8$.

### 6.5. Tradeoff between Spectral Efficiency and Delay Spread

Figure 12 shows the average SE and delay spread for one vehicle pass with different $K$. It can be seen that the case with $K = 16$ obtains the highest SE (i.e., 4.7 bit/s/Hz), but also the largest delay spread (i.e., $2.6 \times 10^{-6}$ s). Increasing $K$ cannot further improve the SE duo to transceiver HWI. This result reveals that when we design a multi-RIS system, the number of RIS $K$ should be carefully considered in terms of the tradeoff between SE and delay spread.

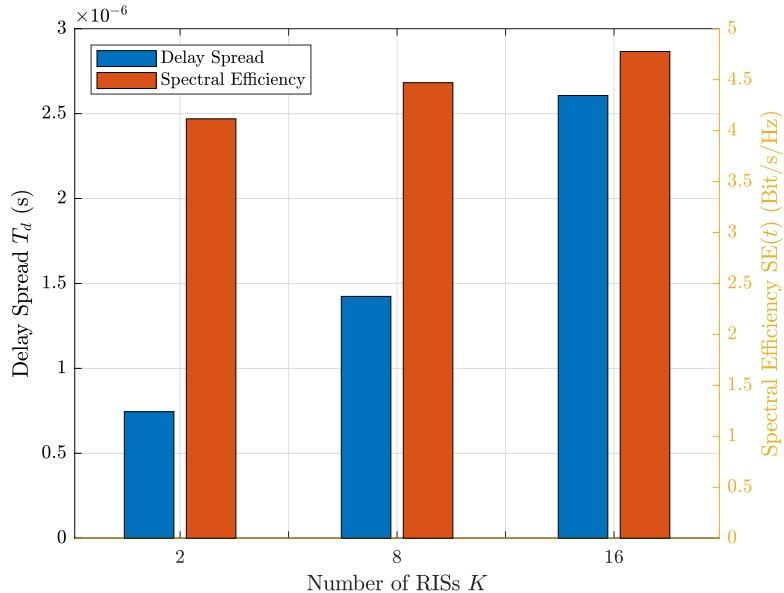

**Figure 12.** Average delay spread and spectral efficiency for different $K$. $a_{\mathrm{HWI}} = \pi/2$, and $\kappa_t = \kappa_r = 0.05^2$.

### 6.6. Study Small-scale Fading for the Proposed Model

Lastly, we extend our LoS channel model (12) and study the effect of small-scale fading on the performance of spectral efficiency. To account for small-scale fading, we assume the Rician fading channel model for all channels involved [8]. Thus, based on (12), the BS to the $(m, n)$-th element of the $i$-th RIS channel with small-scale fading can be expressed as

$$G_{\mathrm{BS}}^{i_m, i_n} = \underbrace{\sqrt{\frac{\beta}{1+\beta}} E_{\mathrm{BS}}^{i_m, i_n}}_{\text{LoS component}} + \underbrace{\sqrt{\frac{1}{1+\beta}} F_{\mathrm{BS}}^{i_m, i_n}}_{\text{NLoS component}}, \tag{45}$$

where

$$E_{\mathrm{LoS}}^{i_m, i_n} = \frac{\lambda_c}{4\pi d_{\mathrm{BS}}^{i_m, i_n}} e^{-j 2\pi f_c \tau_{\mathrm{BS}}^{i_m, i_n}}, \tag{46}$$

with $\tau_{\mathrm{BS}}^{i_m, i_n} = d_{\mathrm{BS}}^{i_m, i_n} / c_0$, and $F_{\mathrm{Rayleigh}}$ are Rayleigh fading components. The BS–vehicle and RIS–vehicle channels are also generated by following a similar procedure. Obviously, the proposed Rician fading model is reduced to the LoS model (12) when $\beta \to \infty$ or Rayleigh fading channel when $\beta = 0$.

In Figure 13, we plot the spectral efficiency for different Rician fading factors $\beta = \{10^9, 5, 3, 1\}$ for one vehicle pass. Note that $10^9$ approximates the LoS model. It can be seen that the spectral efficiency decreases with the diminishing Rician fading factor $\beta$. This is because the reduction of $\beta$ leads to the mismatch of the direct and cascaded links. However, the proposed system with optimal phase shift set $\phi_{i_m, i_n}^{\mathrm{opt}}(t)$ in (33) is still effective when small-scale fading exists.

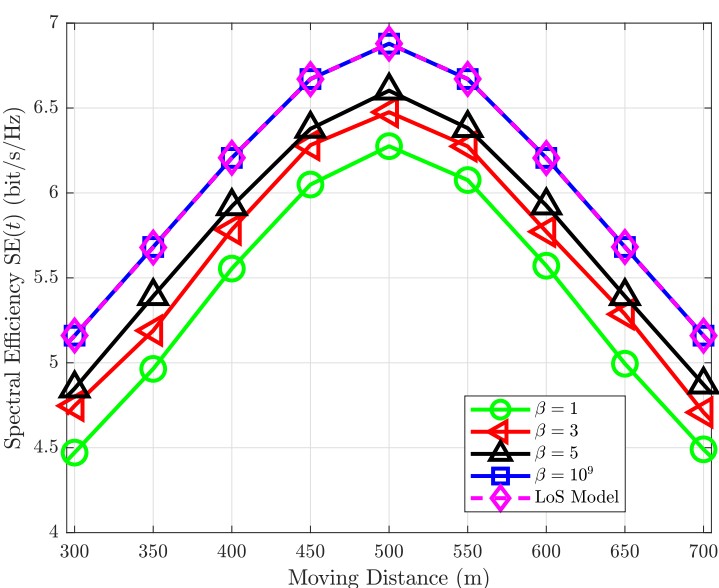

**Figure 13.** Spectral efficiency for different $\beta$. $K = 8$.

## 7. Conclusions and Future Works

In this paper, we have presented an PI-based multi-RIS-assisted system model for HSC with HWI. We have designed and compared different phase shift optimization strategies of the RISs, and obtained a phase shift set that can maximize SE, remove Doppler spread, and keep delay spread at a very low level. Furthermore, we have derived the closed-form expression of SE. We have also compared the performance of different numbers of RISs with HWI in terms of SE and delay spread. The LoS channel model has been extended to the Rician case to show rationality in practice. Numerical evaluations revealed that when the number of RISs increases from 2 to 16, the range of average spectral efficiency and delay spread are from 4 to 4.6 Bit/s/Hz and from 0.7 µs to 2.5 µs, respectively. Therefore, the tradeoff between SE and delay spread should be carefully considered for multi-RIS communication systems. For future works, one promising research direction is using RISs as Doppler generators to obtain Doppler diversity [36,44]. In particular, an RIS can be used not for aligning the direct and cascaded links, but for creating suitable Doppler shifts [12], which can achieve Doppler diversities to enhance the transmission quality. Besides, it is important to optimize the phase shift set under the assumption that the Rician channel exists. Another possible research direction is hardware degradation in RISs. The RIS and transceiver HWI considered in this paper do not relate to usage times. In practice, however, hardware degradation caused by usage time cannot be ignored [45], which is challenging but worth studying in-depth in the future. Moreover, it is necessary to pay attention to the security of the storage of the phase shift set. In this paper, the phase shift set is designed and stored in the BS and is sent to each RIS in real-time. However, centralized storage can result in a single point of failure and security concerns. Distributed storage techniques, therefore, could be utilized to overcome the potential problems [46].

**Author Contributions:** Conceptualization, K.W., C.-T.L. and B.K.N.; formal analysis, K.W.; methodology, K.W., C.-T.L. and B.K.N.; supervision, C.-T.L.; validation, K.W., C.-T.L. and B.K.N.; writing—original draft, K.W.; writing—review & editing, C.-T.L. and B.K.N. All authors have read and agreed to the published version of the manuscript.

**Funding:** This research received no external funding.

**Institutional Review Board Statement:** Not applicable.

**Informed Consent Statement:** Not applicable.

**Data Availability Statement:** Not applicable.

**Conflicts of Interest:** The authors declare no conflict of interest.

## Appendix A

Consider (32) and (33) and omit the HWI; we have

$$T(t) = \max_i \left\{ T_i(t) \right\} = \max \left\{ T_{0i}(t) \right\} \tag{A1}$$
$$= \left\{ \max\{\tau_{i_m,i_n}(t)\} + \frac{\lceil f_c(\max\{\tau_{i_m,i_n}(t)\} - \tau_0(t)) \rceil}{f_c} \right\} - \tau_0(t),$$

and recalling (9), we have

$$\max \left\{ \tau_{i_m,i_n}(t) \right\} = \frac{\max\left\{ d_{\text{BS}}^{i_m,i_n} + d_{i_m,i_n}^{\text{V}}(t) \right\}}{c_0}. \tag{A2}$$

Therefore, ignoring the predictable information error for the trajectory, the RIS that is furthest from the BS, i.e., $\max\{d_{\text{BS}}^{i_m,i_n}\}$, decides the delay spread of the whole system. In other words, if we want to reduce $T(t)$, we have to minimize $\max\{d_{\text{BS}}^{i_m,i_n}\}$. The proof of Proposition 1 is completed.

## Appendix B

We simplify $z(t)$ first; Note that for the signal with unit power, we drop $x(t - \tau(t))$ and $x(t - \tau_{i_m,i_n} - \frac{\phi_{i_m,i_n}(t) + \gamma_{i_m,i_n}(t)}{2\pi f_c})$; then, the (35) can be simplified as

$$z(t) = A_0(t)\sqrt{P_t}\Theta_0(t) + \sum_{i=1}^{K}\sum_{m=1,n=1}^{M,N} A_{i_m,i_n}(t)\sqrt{P_t}\Theta_{i_m,i_n}^{\text{HWI}}(t). \tag{A3}$$

Recall the definitions of $\Theta_0(t)$ and $\Theta_{i_m,i_n}^{\text{HWI}}(t)$; (A3) can be rewritten as

$$z(t) = \sqrt{P_t}\left\{ A_0(t)e^{-j2\pi f_c\tau_0(t)} + \sum_{i=1}^{K}\sum_{m=1,n=1}^{M,N} A_{i_m,i_n}(t)e^{-j2\pi f_c\tau_{i_m,i_n}(t) - j\phi_{i_m,i_n}(t) - j\gamma_{i_m,i_n}} \right\}. \tag{A4}$$

Considering the designed phase shift set $\phi_{i_m,i_n}^{\text{opt}}(t)$ in (33), (A4) can be rewritten as

$$z(t) = \sqrt{P_t}\left\{ \underbrace{A_0(t)e^{-j2\pi f_c\tau_0(t)}}_{z_1(t)} + \underbrace{\sum_{i=1}^{K}\sum_{m=1,n=1}^{M,N} A_{i_m,i_n}(t)e^{-j2\pi f_c\tau_0(t) - j\gamma_{i_m,i_n}}}_{z_2(t)} \right\}. \tag{A5}$$

Next, we know that $\mathbb{E}\{|z(t)|^2\} = P_t\mathbb{E}\{(z_1(t) + z_2(t))^*(z_1(t) + z_2(t))\}$. Then, we have $\mathbb{E}\{(z_1(t) + z_2(t))^*(z_1(t) + z_2(t))\} = \mathbb{E}\{z_1^*(t)z_1(t)\} + \mathbb{E}\{z_2^*(t)z_2(t)\} + \mathbb{E}\{z_1^*(t)z_2(t) + z_2^*(t)z_1(t)\}$. At time $t$, we omit the expression $(t)$ for brevity; then, we have

$$\mathbb{E}\{z_1^*z_1\} = A_0^2, \tag{A6}$$

and

$$\mathbb{E}\{z_2^* z_2\} = \mathbb{E}\left\{ \left| \sum_{i=1}^{K} \sum_{m=1,n=1}^{M,N} A_{i_m,i_n} e^{-\jmath 2\pi f_c \tau_0 - \jmath \gamma_{i_m,i_n}} \right|^2 \right\}$$

$$= \mathbb{E}\left\{ \left| \sum_{i=1}^{K} \sum_{m=1,n=1}^{M,N} A_{i_m,i_n} e^{-\jmath \gamma_{i_m,i_n}} \right|^2 \right\}. \tag{A7}$$

For the *i*-th RIS, let $k = 1, 2, \ldots, MN$; then, we can rewrite (A7) as

$$\mathbb{E}\left\{ \left| \sum_{k=1}^{MN} A_k e^{-\jmath \gamma_j} \right|^2 \right\} = \sum_{k=1}^{MN} A_k^2 + A_1 \left| \sum_{k \neq 1}^{MN} A_k \mathbb{E}\left\{ e^{-\jmath \gamma_1} e^{-\jmath \gamma_k} \right\} \right| +$$

$$\cdots + A_{MN} \left| \sum_{k \neq MN}^{MN} A_k \mathbb{E}\left\{ e^{-\jmath \gamma_{MN}} e^{-\jmath \gamma_j} \right\} \right|. \tag{A8}$$

Because $f(x) = e^{-\jmath x}$ is a continuous function in $\mathbb{R}$, and $\gamma_c$ as well as $\gamma_y$ are independent and identically distributed (i.i.d) random variables, then $|\mathbb{E}\{e^{-\jmath \gamma_x} e^{-\jmath \gamma_y}\}| = |\mathbb{E}\{e^{-\jmath \gamma_x}\}| \cdot |\mathbb{E}\{e^{-\jmath \gamma_y}\}|$. Recall $\gamma \sim \mathcal{U}(-a_{\mathrm{HWI}}, a_{\mathrm{HWI}})$; due to the law of large numbers and the symmetry of the odd function $\sin(\gamma)$, we have $|\mathbb{E}\{e^{-\jmath \gamma}\}|^2 = \mathrm{sinc}^2(a_{\mathrm{HWI}})$, where $\mathrm{sinc}(a_{\mathrm{HWI}}) = \sin(a_{\mathrm{HWI}})/a_{\mathrm{HWI}}$. RIS HWI is i.i.d.; therefore, we have

$$(A6) \approx \sum_{k=1}^{MN} A_k^2 + \mathrm{sinc}^2(a_{\mathrm{HWI}}) \left( A_1 \sum_{k \neq 1}^{MN} A_k + A_2 \sum_{k \neq 2}^{MN} A_k + \cdots + A_{MN} \sum_{k \neq MN}^{MN} A_k \right). \tag{A9}$$

Considering the total *K* RISs, (A7) can be rewritten as

$$\mathbb{E}\{z_2^* z_2\} = \mathrm{sinc}^2(a_{\mathrm{HWI}}) A^\star + \sum_{i=1}^{K} \sum_{m=1,n=1}^{M,N} A_{i_m,i_n}^2, \tag{A10}$$

where $A^\star = A_1 \sum_{k \neq 1}^{KMN} A_k + A_2 \sum_{k \neq 2}^{KMN} A_k + \cdots + A_{KMN} \sum_{k \neq KMN}^{KMN} A_k$.
We then focus on $\mathbb{E}\{z_1^* z_2 + z_2^* z_1\}$. Recalling (33), we have

$$\mathbb{E}\{z_1^* z_2 + z_2^* z_1\} = 2A_0 \mathbb{E}\left\{ \sum_{i=1}^{K} \sum_{m=1,n=1}^{M,N} A_{i_m,i_n} \cos(\gamma_{i_m,i_n}) \right\}. \tag{A11}$$

Since $|\mathbb{E}\{\cos(\gamma_l)\}|^2 = \mathrm{sinc}^2(a_{\mathrm{HWI}})$ [18], (A11) can be rewritten as

$$\mathbb{E}\{z_1^* z_2 + z_2^* z_1\} = 2A_0 \,\mathrm{sinc}(a_{\mathrm{HWI}}) \sum_{i=1}^{K} \sum_{m=1,n=1}^{M,N} A_{i_m,i_n}. \tag{A12}$$

Because $\mathbb{E}\{\log_2(1 + \frac{x}{y})\} \approx \log_2(1 + \frac{\mathbb{E}\{x\}}{\mathbb{E}\{y\}})$ [15] and considering (A6), (A10), and (A12), the proof of Proposition 2 is completed.

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
