# Peer review of "Positioning Information Based High-Speed Communications with Multiple RISs: Doppler Mitigation and Hardware Impairments"

_applsci, doi:10.3390/app12147076_

Round 1

Reviewer 1 Report

In this paper, we consider multiple reconfigurable intelligent surfaces (RISs) assisted systems using positioning information (PI) to explore the potential of Doppler effect mitigation and spectral efficiency (SE) enhancement in high-speed communications (HSC) in the presence of hardware impairments ( HWI). We first present a general multi-RIS-assisted system model for HSC with HWI.

This is actually a comparison with the other works.

They have to show their optimization.

There is no mention of the minimum and maximum distance and angle of radiation.

An algorithm for this work must be shown. 

Figures 13, 11 and 10 should specify the optimization.

Get help from similar articles.

Reviewer 2 Report

1)      Authors should add motivation and contribution in the introduction section

2)      Author should make one table in related study with existing work and their limitations?

3)      How the proposed model is beneficial in decentralized system?

4)      All the table must be improved and the text within the table must be aligned properly.

5)      The grammar and typos error must be taken care 

6)      Author should add advantages and disadvantages of the proposed model.

7)      Author should see the below manuscript and make suitability that how the proposed system model is ok with decentralized structure?

  Towards design and implementation of security and privacy framework for internet of medical things (iomt) by leveraging blockchain and ipfs technology. The Journal of Supercomputing77(8), 7916-7955.

Reviewer 3 Report

The paper presents a multiple reconfigurable intelligent surfaces assisted systems that uses positioning information

- the state of the art is comprehensive, based on 45 titles, most of them very recent

- for clarity reasons, I would avoid writing equations inline with the text (like \omega(M, dx, M) in rows 139-140. It is difficult to follow.

- a list with all the notations used in the equations will be benefic. Some of them are not fully explained in the text, like \tau_{im,in} in eq (9), \gamma_{im’, in’} and \phi_{im’, in’} from eq (15), etc.

- explain in words, the strategies in Table 1 (rows 261-262).

- proof of proposition 1 can be moved in the appendix

- regarding 6. Simulation Results and Discussions – a description of the simulation setup will be beneficial (simulation tool, settings, etc)

- a comparison of the results with similar ones that have been obtained in existing literature is required, to check the truthfulness of the approach and of the obtained results.  

Note: The The Turnitin check reported 19% similarity index, being 5% similar with another paper of the sema authors 

Ke Wang, Chan-Tong Lam, Benjamin K. Ng. "Doppler Effect Mitigation using Reconfigurable Intelligent Surfaces with Hardware Impairments", 2021 IEEE Globecom Workshops (GC Wkshps), 2021.

If the authors can distance themselves from their previous work it will be beneficial. 

Reviewer 4 Report

1. Abstract, summarize the numerical results of proposed work, and discuss how it outperforms existing works.

2. Related work should be mentioned in a separate section by highlighting the comparative analysis in tabular manner. What are the unique features of this study compared to the existing works?

3. A ‘Research Gap’ section should incorporate which will states the purpose of the study

4. Conclusion also required presenting in more quantitative manner

5. Authors reported that the proposed method is novel, but there is no comparison with the existing literature. Comparative analysis is presented only within considered cases. A detailed comparative analysis must be required to support the novelty of the work.

6. How the intelligent term is incorporated in the reconfigurable surfaces are incorporated is not clear in the present form. 

Round 2

Reviewer 1 Report

All changes are added. I accept this manuscript as a paper.

Author Response

Thank you very much for all your comments to further improve the quality of the manuscript.

Reviewer 3 Report

I consider that, after revising the manuscript, the authors improved its quality and now it can be published in a good reputation Journal as Sensors is. 

Author Response

(The authors gave the same response as above.)

Reviewer 4 Report

Authors tried to incorporate most of the comments of reviewer, but still the comparative analysis, presented by the authors, is not sufficient. Reviewer suggest that the comparative analysis should be based on the obtained results compare to the existing methods for the selected system 
